# Endoscopic Ultrasound-Guided Hepaticogastrostomy in Malignant Biliary Obstruction: A Comprehensive Review on Technical Tips and Clinical Outcomes

**DOI:** 10.3390/diagnostics14232644

**Published:** 2024-11-24

**Authors:** Stefano Mazza, Graziella Masciangelo, Aurelio Mauro, Davide Scalvini, Francesca Torello Viera, Marco Bardone, Letizia Veronese, Laura Rovedatti, Simona Agazzi, Elena Strada, Lodovica Pozzi, Chiara Barteselli, Carmelo Sgarlata, Valentina Ravetta, Pietro Fusaroli, Andrea Anderloni

**Affiliations:** 1Gastroenterology and Endoscopy Unit, Fondazione IRCCS Policlinico San Matteo, 27100 Pavia, Italy; 2Gastroenterology Unit, Hospital of Imola, University of Bologna, 40026 Imola, Italy; 3Department of Internal Medicine and Medical Therapeutics, University of Pavia, 27100 Pavia, Italy

**Keywords:** endoscopic ultrasound, hepaticogastrostomy, biliary drainage, malignant biliary obstruction, efficacy, safety

## Abstract

Endoscopic ultrasound-guided biliary drainage (EUS-BD) has dramatically spread and improved in the last two decades and is changing the paradigm of drainage in case of malignant biliary obstruction (MBO). EUS-BD can be achieved from different routes, including the common bile duct (choledochoduodenostomy), intrahepatic bile ducts (hepaticogastrostomy), and gallbladder as a rescue (cholecystogastrostomy/cholecystoduodenostomy). EUS-guided hepaticogastrostomy (EUS-HGS) is a valuable option for biliary drainage in MBO when ERCP fails or is not feasible. EUS-HGS has demonstrated high efficacy with a good rate of technical and clinical success. The safety profile is also overall favorable, although severe adverse events may occur in a significant proportion of patients. From a technical perspective, EUS-HGS is considered one of the most demanding procedures in biliopancreatic endoscopy, requiring multiple steps and high technical skills and experience. In this comprehensive review, technical tips and clinical outcomes of EUS-HGS are reviewed according to the latest evidence in the literature.

## 1. Introduction

Endoscopic retrograde cholangiopancreatography (ERCP) is currently considered the first-line approach for the treatment of malignant biliary obstruction (MBO). However, ERCP may fail due to unsuccessful cannulation or unreachable papilla, and percutaneous transhepatic biliary drainage (PTBD) was classically considered the alternative. However, PTBD is burdened with high morbidity and mortality, which greatly affect patients’ quality of life. Since its first introduction in 2001, EUS-BD has been expanding considerably as a valuable procedure to treat biliary obstruction, and has become the second-line procedure of choice after ERCP failure because of its favorable safety profile compared to PTBD [1,2].

In 2003, Giovannini et al. firstly described a case of palliative biliary drainage in a patient with inoperable hepatic hilar obstruction, performed under endoscopic ultrasound guidance by creating an anastomosis between the dilated left hepatic duct and the stomach [3]. In a subsequent pilot study on 11 patients with biliary hilar obstruction and failed ERCP due to surgically altered anatomy or duodenal stenosis, endoscopic ultrasound-guided hepaticogastrostomy (EUS-HGS) was successfully performed in 10/11 cases, with clinical success in all patients [4].

Nowadays, EUS-HGS and EUS-guided choledochoduodenostomy (EUS-CDS), along with EUS-guided gallbladder drainage as a rescue, are the main EUS-BD procedures performed in case of MBO after ERCP failure. This review will focus on technical features and clinical outcomes of EUS-HGS in the setting of MBO, including a comparison with other biliary drainage techniques, according to the most recent evidence in the literature.

## 2. Current Indication for EUS-Guided HGS in Malignant Biliary Obstruction

ERCP is currently the first-line technique for biliary drainage in MBO, both distal and proximal/hilar. However, ERCP can fail or not be feasible as in cases of failed biliary cannulation, duodenal stenosis, or surgically altered anatomy.

In distal MBO after ERCP failure, different options are possible, with EUS-CDS and EUS-HGS being the best in terms of efficacy and safety; they are currently suggested instead of PTBD in this setting [5,6,7,8]. Regarding what should come first between these two techniques, current guidelines recommend EUS-CDS over EUS-HGS when the former is feasible, because of its better safety profile [5,8]. Moreover, if evidence on EUS-CDS as an alternative to ERCP for primary drainage in distal MBO is growing [9,10,11], data on EUS-HGS as first-line option are very limited. However, EUS-CDS may be not feasible if the common bile duct is not dilated enough, if the duodenal wall is infiltrated by the tumor, or if the duodenum is not accessible at all because of stenosis or altered anatomy; in these cases, EUS-HGS is considered the treatment of choice providing that left intrahepatic bile ducts are sufficiently dilated [5,8].

If the biliary obstruction is located at the proximal/hilar portion, EUS-HGS is a valuable option both as second choice after failed ERCP, or in combination with ERCP when drainage of both liver lobes is warranted [5,7]. Indeed, at least 50% of liver parenchyma should be drained in case of Bismuth types II–IV hilar strictures, but this may not be possible using ERCP alone when left and right biliary systems are disconnected and transpapillary access to one of the two systems is impossible [12,13]. In this case, PTBD has classically been performed to enhance a complete drainage, however the utility of EUS-HGS in combination with ERCP in this setting is increasingly being reported [14,15,16].

## 3. Technical Aspects

### 3.1. Preoperative Evaluations

As stated above, EUS-HGS is challenging and technically demanding. Pre-operative evaluation is therefore fundamental to approach the procedure with as much awareness as possible, maximizing efficacy while minimizing risk of complications. When considering EUS-HGS for any indications, a multidisciplinary discussion involving at least an endoscopist and a surgeon, with other specialists like an interventional radiologist, oncologist and anesthesiologist as required, is essential to share the management plan and to guarantee rescue/conversion strategies in case of failure or adverse events.

Available abdominal imaging, particularly CT scan, should be carefully evaluated for the presence of features that may foresee a more difficult and risky procedure, particularly intrahepatic bile ducts dilation, site of the biliary obstruction, peri-hepatic ascites, liver masses, relation between stomach and liver, collateral vessels around the stomach. Significant ascites located between stomach and liver has been considered a contraindication to EUS-HGS since it prevents fistula maturation and increases the risk of bile leakage, bleeding and stent migration [17,18,19].

According to current guidelines, EUS-HGS should be regarded as a high-risk bleeding procedure, therefore anticoagulant therapy should be discontinued whenever possible. However, initial data seem to exclude an increased risk of adverse events in patients on antiplatelet and/or anticoagulant treatment who undergo EUS-BD, including HGS [20].

Even though no studies have demonstrated a positive impact of antibiotic prophylaxis on clinical outcomes in patients undergoing EUS-HGS, some studies reported routine antibiotic dose administration prior or just after the procedure, and such behavior is suggested by main international guidelines on therapeutic EUS [5,6,8].

The procedure must be performed in an endoscopic room equipped with fluoroscopy and with all the devices for advanced biliopancreatic procedures, according to current guidelines [17]; operators should be experienced with both advanced EUS and ERCP.

The patient’s position is an important aspect to consider, since it must ensure a precise orientation of left–right and proximal–distal directions on the fluoroscopy screen, as well as a good panoramic view of the intrahepatic biliary tree. The latter is especially important when approaching hilar obstructions, since an accurate evaluation of the hepatic ducts involvement and the maintenance or not of communication between the two biliary hemi-systems is of paramount importance. We therefore suggest a supine position when performing EUS-HGS.

The procedure is generally performed under deep sedation or general anesthesia, depending on the patient’s condition and at the discretion of the anesthesiologist. This agrees with recent guidelines by the British Society of Gastroenterology on sedation in gastrointestinal endoscopy, which suggest therapeutic EUS be performed under deep sedation or general anesthesia [21]. No comparative data are to date available suggesting one sedation better than another; therefore, the definitive choice should be made based on local protocols and expertise.

Notably, informed consent should be goal-based and no longer procedure-based, since switching to different endoscopic procedures to achieve adequate biliary drainage may be necessary depending on technical and intraprocedural findings, with the aim of obtaining drainage in the most effective and safe way [7,22].

### 3.2. Intraprocedural Aspects

EUS-HGS is a complex procedure that involves multiple steps, each with particular challenges and deserving of specific consideration. The main steps are the following:Intrahepatic bile duct punctureFluoroscopy and guidewire manipulation into the biliary treeTract dilationStent positioning

In the standard EUS-HGS technique (i.e., in case of normal anatomy), the left lateral branch of the intrahepatic bile duct is punctured from the stomach using a fine-needle aspiration (FNA) needle. Both B2 (lateral posterior branch) and B3 (lateral anterior branch) segments can be punctured, without definitive evidence on which access site is more suitable for EUS-HGS. A recent multicenter, retrospective study enrolling 161 patients explored procedural outcomes by dividing between the B2 and B3 puncture sites. The authors found similar technical success, clinical success, and adverse events rate, with a short procedural time for the B2 group [23]. In general, a dilated B2 segment is easier to be punctured and for guidewire manipulation since it runs straighter in relation to scope and needle direction; however, B2 has a more cranial position leading to the risk of a transesophageal or transcardial puncture, which may be related to severe adverse events, such as mediastinitis [24,25]. On the other side, for B3 access the tip of the scope is almost surely located in the stomach, but it is more flexed (around 90°) making position less comfortable for needle puncture, guidewire manipulation, and stent release. The angle between the needle and the scope is also important, since Ogura et al. showed that an angle >135° is associated with successful guidewire insertion [26]. When choosing the puncture site, the distance from the scope and the grade of bile duct dilation should also be taken into account; indeed, Oh et al. demonstrated that targeting a bile duct with a diameter > 5 mm and at a distance ≤ 3 cm from the probe significantly increases technical success [27], however this must be balanced with the findings by Yamamoto et al., who showed that a distance of less than 2.5 cm between the liver surface and the bile duct to be an independent risk factor for biliary peritonitis [28]. Generally, the scope position and the puncture site should be carefully evaluated both at fluoroscopy and EUS view, since selecting the best point helps maximize the efficacy and safety of the procedure. Examples of fluoroscopy images of the EUS scope position when puncturing B2 and B3 segments are shown in Figure 1.

A 19-gauge needle is usually used for the puncture, with a 0.035-inch or a 0.025-inch guidewire passing through. The use of 22-gauge needle with a 0.018-inch guidewire has been reported to be feasible and may be useful in case of slight dilation of the bile duct [29], although the lower force transmission of the 0.018-inch guidewire must be considered and thicker guidewires should be preferred whenever possible. In a recent retrospective series of 14 patients undergoing EUS-HGS with 22-gauge FNA needle and 0.018-inch guidewire, technical success was achieved in 11 patients, but switch to a 0.025-inch guidewire was necessary in 8 cases [30]. When accessing the bile duct with the needle, it is also important to avoid vessel puncture by applying ColorDoppler and maintaining ultrasound guidance.

Once access to the bile duct is obtained with the needle, contrast medium is injected to confirm the position and to assess the biliary tree anatomy, particularly at the liver hilum (Figure 1); the amount of contrast should be limited to what is strictly necessary to avoid increase in biliary pressure. The guidewire is then passed through the needle and manipulated inside the bile ducts, in what is considered one of the most challenging steps of this procedure (Figure 2). Ideally, the guidewire should be advanced beyond the stenosis and across the papilla (or the biliary-enteric anastomosis in case of altered anatomy) to the duodenal lumen; this situation provides the greatest stability for subsequent exchanges and advancement of the various devices. When this is not possible, as in the case of particularly complex biliary stricture or anatomy, the wire should, however, be advanced deeply into the common bile duct or into the biliary tree, to ensure enough stability and pulling force for the advancement of subsequent devices. The choice of the wire is important and should be tailored by the endoscopist based on the biliary anatomy and position of the scope, obtaining the optimal balance between flexibility of the tip, which aids wire progression, and stiffness, which helps during stent insertion. If the guidewire goes toward the liver periphery, it must be pulled back for repositioning; this step may be critical because of the risk of wire shearing. For this reason, new needles have been designed for interventional EUS. One is the EchoTip Access Needle by Cook Medical (Bloomington, IN, USA) which is characterized by a sharp stylet that has to be fully inserted for puncture, and a needle tip that becomes blunt when the stylet is removed, thus avoiding guidewire shearing [31]. Another one is the Beacon EUS Access System by Covidien/Medtronic, which again has a sharp stylet to allow for puncture and a blunt-tipped needle, but in addition the tip assumes a predetermined curvature (90° or 135°) after stylet removal and is fully rotatable, allowing the wire to be coaxial to the needle tip during manipulation [32]. Experience with these needles is, however, limited and they are so far poorly available. To reduce the risk of needle shearing with standard FNA needles, the so called “liver impaction technique” has been proposed by Ogura et al., which consists of withdrawing only the needle within the liver parenchyma before pulling back the guidewire; this may prevent wire shearing thanks to both a tamponade effect of liver parenchyma around the needle tip, and an increase in the angle between the needle and the wire [33].

After the guidewire has been properly and steadily inserted into the biliary tree, the access tract needs to be dilated in order to create the fistula and allow the passage of devices with larger catheters, such as stent catheters. Fistula tract dilation may be obtained using mechanical or electrocautery devices. For mechanical dilation, a tapered dilator or a hydrostatic ballon can be used. Several types of dilators have been studied for EUS-HGS, including ultra-tapered, ultra-thin, and drill dilators [34,35,36,37]. For ballon dilation, usually a 4 mm or 3 mm ballon is used, allowing for a larger tract dilation compared to bougie or electrocautery dilators, but potentially with higher risk of bile leakage [19]. Electrocautery devices include cystotome, usually 6-Fr, or needle-knife; both can be advanced over the wire, but while the tip of the cystotome and the guidewire are coaxial, with the needle-knife the axis of the needle is misaligned with the guidewire, leading to an increased risk of damage to the surrounding tissue and bleeding. In their prospective study, Park et al. found that the use of needle-knife was independently associated with postprocedural adverse events, while mechanical dilation showed no correlation [38]. Similar results emerged in the comparative trial (EUS-HGS vs. EUS-CDS) by Khashab et al., where a comparison between a coaxial and non-coaxial device is made, leading the authors to conclude that non-coaxial electrocautery devices should be avoided if possible because of increased risk of adverse events [39]. Another retrospective study demonstrated a higher risk of bleeding with cautery dilation compared to mechanical dilation, although this complication occurred only when plastic stent was placed [34]. European Society of Gastrointestinal Endoscopy (ESGE) guidelines suggests fistulous tract be created using a 6-Fr cystotome or alternatively by mechanical dilation; due to its high penetrating ability and relatively small caliber, the 6-Fr cystotome is considered an ideal accessory that allows the introduction of various tools and stent-introducing catheters, without leading to clinically significant bile leakage or pneumoperitoneum [17,40]. Examples of fluoroscopy images of fistulous tract dilation with 6-Fr cystotome and 4 mm dilation balloon are shown in Figure 3.

As for stent choice, self-expandable metal stents (SEMS) are currently widely preferred over plastic stents, since the latter have been associated with increased risk of adverse events and late obstruction [39]. Uncovered SEMS carries an unacceptable risk of bile leakage, whereas fully covered SEMS may occlude intrahepatic bile duct branches and are more prone to stent migration with potentially severe consequences [41]. Thus, dedicated partially covered SEMS (PC-SEMS) have been designed for EUS-HGS. The main features of these stents are the presence of an uncovered portion meant to be deployed inside the bile duct, a covered portion that should cover the fistula reducing the risk of leakage/perforation, and various anti-migration system including flared shape of the proximal end and anchoring flaps. One of the most common dedicated stent for EUS-HGS is the Giobor stent by Taewoong Medical (Seoul, Republic of Korea), which is a partially covered stent designed with a 30% distal uncovered portion and a 70% covered portion, an anti-migration flared proximal end, and radiopaque markers at both stent ends and at the end of the covered part [42,43]. Another dedicated stent is the Hanarostent BPE by M.I. Tech (Seoul, Republic of Korea), which has a flared distal (gastric) portion with a diameter of 20 mm to prevent migration toward the liver, and a 2 cm uncovered proximal (hepatic) end with multiple anti-migratory flaps to ensure stent anchor inside the liver [44]. The main dedicated SEMS currently available for EUS-HGS are summarized in Table 1.

Stent length is also important, since it must ensure enough length within the bile duct to prevent proximal migration, enough length inside the stomach to prevent distal migration, and adequate coverage of the hepatogastric fistula with the covered stent portion to prevent bile leakage. For this reason, 8 cm to 12 cm stents are usually used for EUS-HGS [45,46]. Moreover, an intragastric portion of at least 3 cm has been independently associated with longer stent patency over time [46]. Besides stent characteristics, some tips might be useful to minimize the risk of stent misdeployment, such as the intra-channel release technique [47,48,49], which is indeed suggested by current guidelines on EUS-BD [17,45]. This technique is based on pushing the proximal stent end, which has already been released inside the working channel, towards the gastric wall, while gently leaving the up angle of the scope tip; this movements combination will push the gastric wall forward and the scope away from the gastric wall, therefore maintaining the stomach closer to the liver while ensuring a safe release of the proximal stent portion inside the stomach. Examples of fluoroscopy images of a dedicated partially covered stent (i.e., Giobor stent), both at fluoroscopy and endoscopic view, are shown in Figure 4.

It is worth considering separately the specific situation in which left and right hepatic ducts are disconnected (e.g., in Bismuth types III and IV biliary tumors), or when only the right biliary system is obstructed. To achieve adequate drainage in these cases, two techniques have been described. One involves puncturing the left hepatic duct as in the standard EUS-HGS technique, advancing the guidewire into the right biliary system, and deploying an uncovered SEMS between right and left intrahepatic bile ducts (so called “bridging technique”). A further stent is eventually deployed across the hepatico-gastro-anastomosis as in the standard technique. The other involves directly puncturing the dilated right intrahepatic bile ducts from the duodenal bulb or the antrum, with further steps resembling those of the standard technique, leading to HGS or a hepaticoduodenostomy creation. Despite limited data in the literature, both these techniques proved to be feasible and effective in draining complex biliary structures [50,51,52]; however, it is clear that they add difficulties to the already challenging standard EUS-HGS procedure, and therefore should be reserved for very skilled and experienced operators.

## 4. Efficacy of EUS-Guided HGS

The main evidence on EUS-HGS efficacy is detailed in Table 2; specifically, we considered randomized trials, prospective studies and retrospective studies with at least 30 patients enrolled.

Five randomized trials to date included patients treated with EUS-HGS for MBO after failed ERCP. A PC-SEMS was used in all the three studies. Technical success rate ranged from 87.5% to 100%, and clinical success rate from 80% to 100% [53,54,55,56,57].

In a recent meta-regression analysis, Binda et al. evaluated efficacy and safety of EUS-HGS for the drainage of malignant or benign biliary obstruction. Thirty-three studies were finally included, with a total of 1644 patients. Malignant causes were by far the most common (99.6%), duodenal/papillary invasion being the most frequently reported. Pooled technical and clinical success were 97.7% and 88.1%, respectively. They also found that technical success was higher when considering centers with greater experience (>4 cases/year), and that both technical and clinical success significantly improved passing from studies before 2015 to those after 2015 [58]. These data confirm that experience and expertise are required for this procedure, and technique optimization along with the introduction of dedicated devices may improve its efficacy over time.

Another meta-analysis published the same year showed similar results (94.4% technical success and 88.6% clinical success). Interestingly, the authors performed a subgroup analysis for EUS-HGS with concomitant antegrade stenting, showing technical and clinical success rate of 89.7% and 92.5%, respectively [59]; the rationale is that the double route of drainage could improve the drainage efficacy, although the technical challenges of deploying the transpapillary stent should be considered.

Four prospective studies evaluated the efficacy of EUS-HGS with the use of dedicated stents; range of technical and clinical success rate was 94–100% and 72–95%, respectively [41,44,60,61]. No comparative studies on dedicated vs. non-dedicated stents for EUS-HGS are available to date, thus it is not clear if the introduction of such devices has significantly improved the efficacy of the procedure. However, the use of dedicated stents may reduce the risk of migration, potentially preventing severe consequences such as bile leakage, and might prolong stent patency, which is of paramount importance for patients undergoing oncological therapies and for patients’ quality of life.

In a very recent multicenter prospective study on 20 patients with MBO who failed ERCP, Vargas-Madrigal et al. assessed efficacy and safety of a dedicated cautery-enhanced tubular SEMS (Niti-S Hot Giobor, Taewoong Medical, Seoul, Republic of Korea). They reported a technical and clinical success of 100%, without occurrence of severe adverse events or death; they also reported a very short median procedural time of 16 min (range 6–25) [62]. Like what has been observed for electrocautery-enhanced lumen-apposing metal stents (EC-LAMS), the application of single-step devices in HGS could improve the feasibility of the technique by avoiding the need for multiple device exchanges and increasing the velocity of the maneuver.

**Table 2 diagnostics-14-02644-t002:** Main evidence on EUS-HGS efficacy.

First Author Year	Design	No. of Patients	Condition	Obstruction Site	Stent	Technical Success %	Clinical Success %
Artifon 2015 [53]	RCT	49	Malignant	Distal	PC-SEMS	96	91
Park 2015 [56]	RCT	32	Malignant	Proximal/Distal	Dedicated SEMS	199	94
Paik 2018 [57]	RCT	64	Malignant	Proximal/Distal	Dedicated SEMS	97	81
Minaga 2019 [54]	RCT	47	Malignant	Distal	Dedicated SEMS	88	100
Marx 2022 [55]	Randomized phase II trial	35	Malignant/Benign	Proximal/Distal	Dedicated SEMS	94	80
Moryoussef 2017 [61]	Prospective	18	Malignant	Proximal	SEMS	94	72
Okuno 2018 [41]	Prospective	20	Malignant	Distal	SEMS	100	95
Jagielski 2021 [60]	Prospective	53	Malignant	Proximal/Distal	Dedicated SEMS	98	87
Anderloni 2022 [44]	Prospective	22	Malignant	Proximal/Distal	Dedicated SEMS	100	91
Vila 2012 [63]	Retrospective	34	Malignant	Proximal/Distal	Not specified	65	Not assessed
Poincloux 2015 [64]	Retrospective	66	Malignant/Benign	Proximal/Distal	Plastic, SEMS, Dedicated SEMS	98	94
Khashab 2016 [39]	Retrospective	61	Malignant/Benign	Distal	Plastic, SEMS	92	82
Nakai 2016 [43]	Retrospective	33	Malignant	Proximal/Distal	Dedicated SEMS	100	100
Sportes 2017 [65]	Retrospective	31	Malignant	Proximal/Distal	SEMS	100	86
Oh 2017 [27]	Retrospective	129	Malignant/Benign	Proximal/Distal	Plastic, Dedicated SEMS	93	81
Honjo 2018 [34]	Retrospective	49	Malignant/Benign	Proximal/Distal	Plastic, SEMS	100	Not assessed
Miyano 2018 [48]	Retrospective	41	Malignant/Benign	Proximal/Distal	SEMS	100	100
Nakai 2020 [42]	Retrospective	110	Malignant	Proximal/Distal	Dedicated SEMS	100	94

PC, partially covered; RCT, randomized controlled trial; SEMS, self-expandable metal stent.

## 5. Safety of EUS-Guided HGS

Main evidence on EUS-HGS is summarized in Table 3.

EUS-HGS is a valuable technique for biliary drainage in patients with MBO, particularly when ERCP is not feasible or fails. EUS-HGS is technically challenging and is still performed only by very experienced operators and in advanced endoscopy referral centers. Common HUS-HGS adverse events include abdominal pain, pneumoperitoneum, bile leak, cholangitis/sepsis, bleeding, stent migration, recurrent biliary obstruction. Among these, recurrent biliary obstruction, self-limited pneumoperitoneum, bile peritonitis and bleeding are the most frequent; perforation, intraperitoneal migration of the stent, and mediastinitis are rare but life-threatening.

Two recent comprehensive reviews and meta-analyses reported an overall incidence of AEs of 23.8% and 19%, respectively [69,70]. Another recent meta-analysis reported the rate of overall AEs of 15.5%, serious AEs of 0.6%, and a procedural mortality of 0.2%; interestingly, the incidence of bile leak, bleeding, and stent occlusion was higher for EUS-HGS performed with plastic stents compared with metal stents [71].

Furthermore, an increased distance between the hepatic parenchyma and stomach has been shown to increase bile leak and stent migration [67].

The use of dedicated PC-SEMS has enhanced procedural safety by preventing bile leakage and reducing the risk of stent migration. In a recent prospective, a single-arm study assessing feasibility and safety of the aforementioned dedicated PC-SEMS, the Hanarostent BPE (M.I. Tech), no severe AEs were reported, underlining the efficacy of anti-migratory systems in preventing severe AEs such as perforation, bile leak, and peritonitis, which are mainly related to stent misdeployment or migration [44].

More recently, Nakai et al. reported long-term outcomes data of a dedicated PC-SEMS for EUS-HGS in patients with MBO: the adverse event rate was 25%, about half of which were mild transient fever and abdominal pain. The stent used in this study was a modified Giobor stent (Taewoong Medical), designed with a 1 cm uncovered portion on the hepatic side, and a covered, flared portion on the gastric side to prevent dislodgement [42].

The level of obstruction is an important predictor of efficacy and safety of EUS-HFS. Hilar obstruction has been associated with increased risk of overall AEs as well as stent dysfunction over time; specifically, Bismuth classification type IV has been associated with greater procedure complexity, and significantly associated with ineffective drainage [66]. However, a recent retrospective study evaluating the outcomes of patients undergoing EUS-HGS for perihilar cholangiocarcinoma showed that clinical success, in terms of jaundice or cholangitis resolution, can be achieved in a significant proportion of cases [68].

The main EUS-HGS adverse events are described in detail below.

### 5.1. Bleeding

Bleeding is one of the most common complications of EUS-HGS, being reported in about 2–33% of cases [42,56,58,65,68]. In a very recent meta-analysis on efficacy and safety of EUS-HGS, the pooled rate of bleeding was 2.3% and it was the second most common adverse event after cholangitis/sepsis [58].

Bleeding is generally managed endoscopically, with angiographic embolization or surgical intervention required only in rare severe cases [56,57].

### 5.2. Bile Leakage

EUS-HGS requires the exchange of various devices before stent deployment, and these multiple steps may lead to leakage of bile into the abdominal cavity. Incidence of bile leakage after EUS-HGS varies widely among studies, depending on the study type, definition of the adverse event, and type of stent used. In two recent meta-analyses, bile leak was reported in 1.6% [58] and 3% [71] of cases, respectively.

Most cases are self-limiting and managed conservatively with antibiotics; percutaneous drainage or stent repositioning may be needed in more severe, prolonged cases [41,65].

### 5.3. Infection

Infection rates, including transient fever, cholangitis, and hepatic abscess are relatively high, ranging from 3% to 26%, as reported in recent data about the complications of interventional EUS [72].

However, most infectious complications are classified as mild to moderate, and are treatable conservatively with antibiotic therapy.

### 5.4. Stent Migration

The stent can migrate into the gastric lumen, between the gastric wall or into the peritoneal cavity. Migration into the gastric lumen is often treated endoscopically, while distal migration into the peritoneal cavity leads to important bile leakage and can be potentially fatal, requiring surgical treatment [56,57].

Placement of a double pigtail plastic stent through the metal stent may be considered to stabilize the metal stent. As stated above, important technical tips like the intra-channel stent release have been applied to reduced migration rate [48].

### 5.5. Recurrent Biliary Obstruction

Long-term outcomes, including stent patency and the need for re-intervention, are very important for evaluating the safety of EUS-HGS. In their meta-analyses, Binda et al. reported a pooled recurrent biliary obstruction rate of 16.2%, occurring after a median of 165 days [58], while Giri et al. reported a late migration rate of 1.7%, stent occlusion rate of 11%, and reintervention rate of 20.9% [71].

According to some experts, EUS-HGS is supposed to have a longer stent patency compared to ERCP, because of the distance of the stent from the malignant stricture, reducing the risk of tumor ingrowth and overgrowth. However, obstruction by clogging, food, or reactive tissue has been described [73].

## 6. Comparison of EUS-Guided HGS with Other Biliary Drainage Methods

As discussed above, in the case of ERCP failure, several techniques might be considered for biliary decompression, the main being EUS-CDS and percutaneous drainage.

Two randomized controlled trials (RCT) directly compared EUS-HGS and EUS-CDS after ERCP failure in malignant distal biliary obstruction. The first one by Artifon et al. was a single-center study including 25 and 24 patients per arm, respectively. EUS-CDS was performed with the “classic” multi-step technique, and a PC-SEMS was used for both the procedures. They found similar rates of technical and clinical success, adverse events, procedure time, and survival [53]. The other RCT by Minaga et al. was multi-centric and enrolled 24 and 23 patients, respectively. Techniques for EUS-HGS and EUS-CDS were the same, with a dedicated PC-SEMS used for HGS. The technical success rate was higher for EUS-HGS (87.5% vs. 82.6%, *p* = 0.0278); other outcomes including clinical success, adverse events, procedure time, stent patency and survival were similar [54]. It is worth noting that EC-LAMS is currently the most commonly used stent type for EUS-CDS, having made the procedure more reproducible by avoiding multiple exchanges; however, no studies directly comparing EUS-CDS with LAMS and EUS-HGS with dedicated stent are available to date.

Eight meta-analyses comparing EUS-HGS and EUS-CDS have been published to date [11,70,74,75,76,77,78,79]. Among these, the very recent network meta-analysis by Lauri et al. is notable for including only studies with procedures performed as primary approach for biliary drainage of distal MBO. They compared each other’s various biliary drainage techniques, including ERCP, EUS-CDS, EUS-HGS, and PTBD; regarding specifically the EUS-HGS vs. EUS-CDS comparison, no difference emerged in the rate of technical success, clinical success, and adverse events [11]. Interestingly, they differentiated EUS-CDS performed with LAMS or SEMS, and found that EUS-CDS with LAMS was significantly superior to all other techniques, particularly EUS-HGS, for procedural time (mean difference −8.9 min, 95% confidence interval −14.4 to −3.4) [11].

The other meta-analyses mainly focused on second-line procedures after ERCP failure. The most recent one included 18 studies, with 472 EUS-CDS patients and 503 EUS-HGS patients; partially covered and fully covered SEMS were the most commonly used. The two procedures were comparable for technical and clinical success, and procedure-related adverse events rate; EUS-CDS, however, showed a lower rate of recurrent biliary obstruction and lower procedural time [70]. Similar results emerged from other three meta-analyses [74,76,77]. A significant lower rate of adverse events was however found in three meta-analyses [75,78,79]; in one of these, this difference was restricted to early adverse events, which included pneumoperitoneum, bile leakage, bleeding/hematoma, cholangitis, and biliary peritonitis [75]. It should be noted that, apart from the work by Facciorusso et al. that considered only distal malignant obstruction [77], the other meta-analyses do not uniformly report the location of biliary obstruction (i.e., proximal vs. distal) and do not consider separately EUS-HGS performed for distal obstruction, leading to potential biases when compared to EUS-CDS. Indeed, besides the fact that EUS-CDS is not feasible for proximal obstruction treatment, clinical outcomes in proximal strictures drainage are generally worse than those observed in distal obstruction [79]. Moreover, all these meta-analyses do not distinguish between different types of stents used, particularly LAMS for EUS-CDS, and mainly included studies with SEMS used for both the procedures.

To date, only three studies directly compared EUS-HGS and PTBD. One of them was a French multicenter randomized phase II study involving patients with obstructive jaundice after ERCP failure. Technical and clinical success rates were comparable, but the unexpected high 30-day morbidity observed for PTBD led to stop randomization and inclusion in the PTBD arm; a lower risk of reintervention was also seen in patients treated with EUS-HGS [55]. A retrospective study on patients with MBO not exceeding Bismuth stage I and failed ERCP showed similar results [65]. Conversely, a recent propensity score-matched study on patients with both malignant a benign biliary obstruction found a higher clinical success rate for EUS-HGS compared to PTBD (100% vs. 75%), along with a lower adverse event rate, shorter procedure duration, shorter post-procedure length of stay, and fewer reintervention rates for EUS-HGS [80]. It is worth noting that in all these studies, a metal stent (fully covered or partially covered) was used for the hepatico-gastro-anastomosis.

In the meta-analysis by Facciorusso et al., PTBD was non-inferior to EUS-HGS and EUS-CDS in terms of technical/clinical success and adverse events rate; however, a trend towards higher rates of adverse events was observed for PTBD [77]. In the above-mentioned meta-analysis on primary biliary drainage, despite a similar technical and clinical success rate, PTBD showed a higher rate of adverse events compared to all the other drainage techniques (EUS-HGS, EUS-CDS with LAMS or SEMS, ERCP) [11].

No studies are currently available comparing EUS-HGS to EUS-gallbladder drainage or other methods such as EUS-guided antegrade stenting.

## 7. Conclusions and Future Perspectives

In conclusion, EUS-HGS is a feasible, effective, and safe option for biliary drainage in malignant obstruction when ERCP fails.

Its role as an alternative to EUS-CDS, when the latter is feasible, must be better assessed and clarified, needing further prospective evidence directly comparing the two techniques. When EUS-CDS is not viable, as in surgically altered anatomy, EUS-HGS could become the procedure of choice, even though more evidence about efficacy and safety in this setting are desirable to establish this technique more firmly in the clinical practice. The potential role of EUS-HGS as an alternative to ERCP for primary drainage of MBO has not been explored yet, this being a potentially very interesting field to investigate in the near future.

EUS-HGS is currently considered a challenging procedure, and its not insignificant rate of severe adverse events is an important issue that must be addressed. Progression and improvement in both operators’ technical skills and device performance could enhance greater reproducibility and better safety profile of the technique. More structured training programs and introduction of dedicated devices are moving in this direction.

## Figures and Tables

**Figure 1 diagnostics-14-02644-f001:**
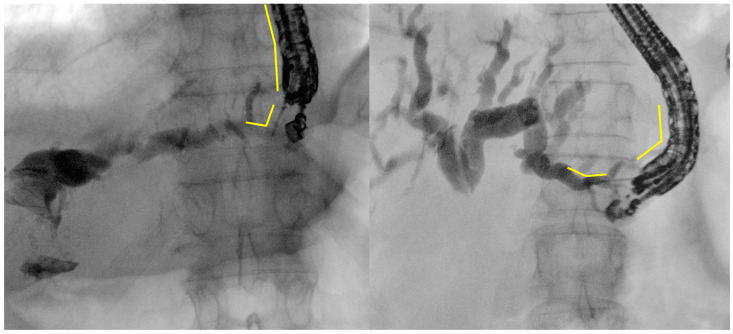
Example images of EUS-guided puncture of intrahepatic bile ducts at the lateral posterior branch (B2, on the **left**) or lateral anterior branch (B3, on the **right**). The difference in the scope tip angulation and position inside the stomach, as well as in the angulation between the needle axis and the bile cut, are clearly visible (yellow lines). These are original figures from the Gastroenterology and Digestive Endoscopy Unit, IRCCS San Matteo Hospital—Pavia, Italy.

**Figure 2 diagnostics-14-02644-f002:**
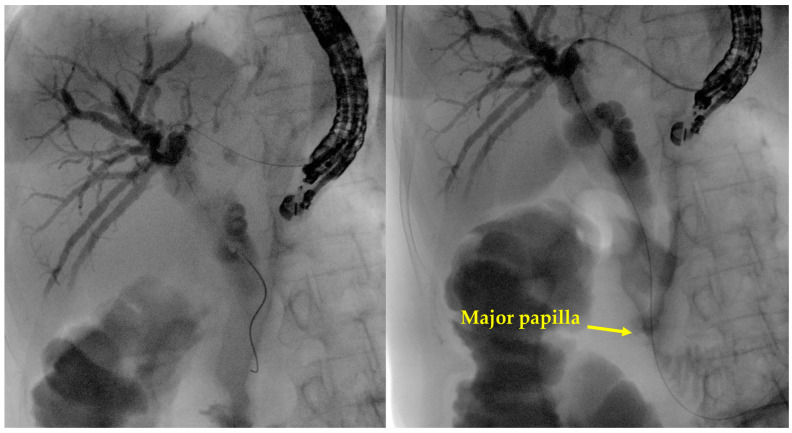
Example images of guidewire manipulation inside the biliary tree during EUS-HGS. On the **left**, the guidewire has been advanced over the hilar stenosis, into the common bile duct. On the **right**, the progression of the guidewire over the papilla, into the duodenal lumen has been obtained. The latter is the best condition for subsequent maneuvers (i.e., dilation and stent positioning) in terms of stability of the wire and pushing force of the devices over the wire. These are original figures from the Gastroenterology and Digestive Endoscopy Unit, IRCCS San Matteo Hospital—Pavia, Italy.

**Figure 3 diagnostics-14-02644-f003:**
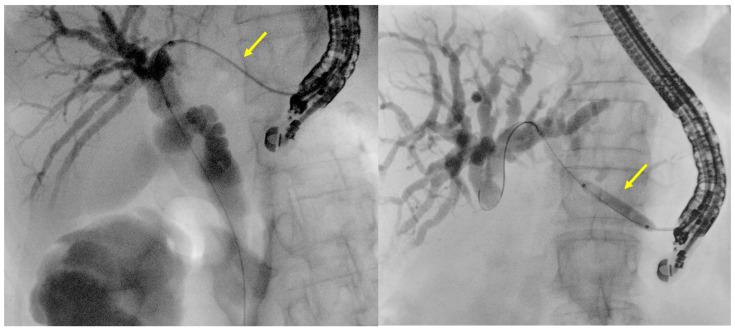
Example images of fistulous tract dilation using a 6-french cystotome (on the **left**, yellow arrow) or a 4 mm balloon on the **right** (yellow arrow). The difference in the degree of dilation achievable is clearly visible. These are original figures from the Gastroenterology and Digestive Endoscopy Unit, IRCCS San Matteo Hospital—Pavia, Italy.

**Figure 4 diagnostics-14-02644-f004:**
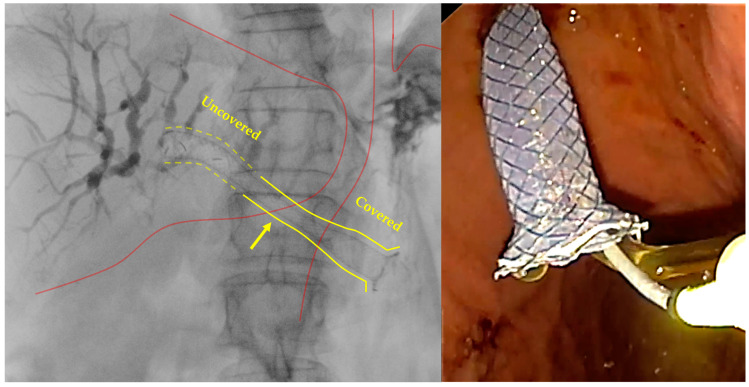
Example images deployed stent during EUS-HGS. On the **left**, a Giobor stent has been deployed (yellow arrow); the different shape of proximal (intrahepatic, dotted yellow lines) and distal (intragastric, yellow continuous lines) portion is clearly visible, as well as the passage between the uncovered portion, which is totally inside the bile duct, and the covered portion, which covers the fistulous tract and goes inside the stomach, are clearly visible. At endoscopic view, the proximal end of the stent released into the stomach is shown, with the flowing bile clearly visible. These are original figures from the Gastroenterology and Digestive Endoscopy Unit, IRCCS San Matteo Hospital—Pavia, Italy.

**Table 1 diagnostics-14-02644-t001:** Main available dedicated metal stents for EUS-HGS.

Stent Name	Company	Available Diameters	Available Lengths	Covering	Anti-Migration Flared Ends	Anti-Migration Flaps	Radiopaque Markers	Electrocautery-Enhanced Catheter
GIOBOR (Niti-S Biliary Covered Stent)	Taewoong Medical, Seoul, Republic of Korea	8, 10 mm	6, 8, 10, 12 cm (for each diameter)	Partially covered 30% Uncovered portion 70% Covered portion	Yes, proximal end	No	Yes, at both stent ends and at the end of covered part	No
HANAROSTENT Biliary	M.I. Tech, Seoul, Republic of Korea	10 mm	4, 5, 6, 7, 8, 9, 10, 11, 12 cm	Partially covered 3 cm Uncovered portion The rest covered	Yes, proximal end	BPD: No BPF: Yes, at both ends	Yes, at both stent ends and at the end of covered part	No
BONASTENT—Hybrid stent	Standard Sci Tech, Seoul, Republic of Korea	8, 10 mm	5, 6, 7, 8, 9, 10 cm (for each diameter)	Partially covered 35 mm Covered portion The rest uncovered	No	Yes, at the covered/uncovered passage, and at proximal end	No	No
BileRush Advance	Piolax Medical Devices, Kanagawa, Japan	8 mm	10, 12 cm	Partially covered 20 mm Uncovered portion The rest covered	Yes, proximal end	No	Yes, at both ends	No
HOT GIOBOR	Taewoong Medical, Seoul, Republic of Korea	8, 10 mm	10 cm	Partially covered 3 cm proximal uncovered portion The rest covered	Yes, proximal end	No	Yes, at both stent ends and at the end of covered part	Yes

**Table 3 diagnostics-14-02644-t003:** Main evidence on EUS-HGS safety.

First Author Year	Design	No. of Patients	Condition	Obstruction Site	Stent	Overall AEs %	Profile of AEs
Park 2015 [56]	RCT	32	Malignant	Proximal/Distal	Dedicated SEMS	25	Cholangitis (1/16) Leak (1/16) Bleeding (1/16) Pneumoperitoneum (4/16)
Paik 2018 [57]	RCT	64	Malignant	Proximal/Distal	SEMS	10	Pneumoperitoneum (2/7) Leak (1/7) Cholangitis (4/7)
Okuno 2018 [41]	Prospective	20	Malignant	Distal	SEMS	15	Cholangitis (2/20) Migration (6/20) Occlusion (4/20)
Sportes 2017 [65]	Retrospective	31	Malignant	Proximal/Distal	SEMS	16	Cholangitis (2/16) Bleeding (1/16) Leak (2/16) Mortality (2/16)
Miyano 2018 [48]	Prospective	41	Malignant	Proximal/Distal	SEMS	40	Leak (3/20) Cholangitis (1/20) Migration (1/20)
Minaga 2017 [66]	Retrospective	30	Malignant	Proximal	SEMS-Plastic	33 (10)	Leak (3/10) Cholangitis (7/10)
Ochiai 2021 [67]	Retrospective	48	Malignant	Proximal/Distal	SEMS	17	Migration (5/8) Leak (3/8)
Anderloni 2022 [44]	Prospective	22	Malignant	Proximal/Distal	Dedicated SEMS	13	Hepatic abscess (3/3)
Nakai 2020 [42]	Retrospective	111	Malignant	Proximal/Distal	Dedicated SEMS	25	Leak (4/27) Cholangitis (3/27) Bleeding (1/27)
Schoch 2022 [68]	Retrospective	34	Malignant	Proximal	SEMS	26	Cholangitis (5/9) Bleeding (3/9) Leak (1/9) Mortality (1/9)

AEs, adverse events; RCT, randomized controlled trial; SEMS, self-expandable metal stent.

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
