# Peer review of "Endoscopic Ultrasound-Guided Hepaticogastrostomy in Malignant Biliary Obstruction: A Comprehensive Review on Technical Tips and Clinical Outcomes"

_diagnostics, 2024, doi:10.3390/diagnostics14232644_

Round 1

Reviewer 1 Report

Comments and Suggestions for Authors

This is a narrative review article regarding EUS-guided hepaticogastrostomy using recent published evidence focusing on its technical issues.

Although this article is well-written and likely to deserve for publication, there are some minor issues to reconsider/

1. Reference 2

Article title is incorrect.

2. Reference 7

Please add journal number and pages

3. Line 132-134

Please add the cited article for this study.

4. Line 142-143

Ogura et al showed that an angle smaller than 135°is associated with successful guidewire insertion, not over 135°. Please confirm that.

5. Line 143-146

Yamamoto et al. showed shorter distance between the scope and bile duct than 2.5cm is the risk factor of bile peritonitis. How do you think about that?

6. Line 214-215

if the tip of the cystotome is coaxial, the axis of the needle is misaligned with the guidewire, leading to an increased risk of damage to the surrounding tissue and bleeding This sentencedoes not make sense.

Author Response

Reviewer 1

This is a narrative review article regarding EUS-guided hepaticogastrostomy using recent published evidence focusing on its technical issues.

Although this article is well-written and likely to deserve for publication, there are some minor issues to reconsider/

1. Reference 2

Article title is incorrect.

Thank you. This reference has been corrected.

2. Reference 7

Please add journal number and pages

Thank you. This reference has been corrected.

3. Line 132-134

Please add the cited article for this study.

Thank you for pointing it out and sorry for missing it. The relevant study has now been cited.

4. Line 142-143

Ogura et al showed that an angle smaller than 135°is associated with successful guidewire insertion, not over 135°. Please confirm that.

Thank you for asking this clarification. If we understand well, according to the cited study by Ogura et al (Ogura T, Nishioka N, Ueno S, Yamada T, Yamada M, Imoto A, Hakoda A, Higuchi K. Effect of echoendoscope angle on success of guidewire manipulation during endoscopic ultrasound-guided hepaticogastrostomy. Endoscopy. 2021 Apr;53(4):369-375. doi: 10.1055/a-1199-5418), an angle between the needle and the scope greater than 135° was associated with successful guidewire manipulation; this could be explained by the fact that with a 135° angle or greater, the scope tip is less angled and the guidewire can therefore be maneuvered more easily.

5. Line 143-146

Yamamoto et al. showed shorter distance between the scope and bile duct than 2.5cm is the risk factor of bile peritonitis. How do you think about that?

Thank you very much for this sharp observation. We think that a short distance between probe and bile duct may increase technical success, due to an easier puncture, but the fewer hepatic parenchyma in between may increase the risk of bile leakage. We have added this important evidence in the revised version, following your suggestion. 

6. Line 214-215

“if the tip of the cystotome is coaxial, the axis of the needle is misaligned with the guidewire, leading to an increased risk of damage to the surrounding tissue and bleeding” This sentencedoes not make sense.

Thank you for underlining this. The sentence has been corrected to “while the tip of the cystotome and the guidewire are coaxial, with the needle-knife the axis of the needle is misaligned with the guidewire, leading to an increased risk of damage to the surrounding tissue and bleeding”.

Reviewer 2 Report

Comments and Suggestions for Authors

Dear Authors 

I had the pleasure to read your thorough narrative review of the existing literature on this interesting topic among endoscopists. The overview is clear even though the english could be improved. The radiological images are quite clear while the endoscopic image of the stent could be improved in quality. As a suggestion, given the workload necessary to collect all the evidence and summarizing it it could be done in a systematic fashion, just to extrapolate further evidence from it

Comments on the Quality of English Language

English could be improved especially in readibility

Author Response

Reviewer 2

Dear Authors

I had the pleasure to read your thorough narrative review of the existing literature on this interesting topic among endoscopists. The overview is clear even though the english could be improved.

Thank you very much for the comments. We have entirely reviewed the english to improve paper readability.

The radiological images are quite clear while the endoscopic image of the stent could be improved in quality.

Thank you, we have improved the quality of this image accordingly.

As a suggestion, given the workload necessary to collect all the evidence and summarizing it it could be done in a systematic fashion, just to extrapolate further evidence from it

We fully agree on the large workload required to collect and summarize this big amount of data, and we really thank you for the suggestion. We have produced a comprehensive review focusing on the most important and recent evidence in the literature, with the intention of combining a more strictly scientific part with one of practical utility. We also aimed to provide a more detailed overview of the main literature evidence in the tables.

Reviewer 3 Report

Comments and Suggestions for Authors

This is a thorough and complete review of utility and safety of EUS-HGS  in the management of malignant biliary obstruction. The information in this manuscript is abundant enough and could be accepted for consideration for publication in the current status. 

Only one question elicited on EUS-CDS and EUS-HGS.  In the situation of distal obstruction, both EUS-CDS and EUS-HGS are helpful. However, in proximal obstruction, EUS-CDS can not play a role. The standard for comparison is not even. It will be better if the authors could have more comments on this issue.  If no further discussion could be added, just let it be.

Author Response

Reviewer 3

This is a thorough and complete review of utility and safety of EUS-HGS in the management of malignant biliary obstruction. The information in this manuscript is abundant enough and could be accepted for consideration for publication in the current status.

Only one question elicited on EUS-CDS and EUS-HGS.  In the situation of distal obstruction, both EUS-CDS and EUS-HGS are helpful. However, in proximal obstruction, EUS-CDS can not play a role. The standard for comparison is not even. It will be better if the authors could have more comments on this issue.  If no further discussion could be added, just let it be.

Thank you very much, I fully agree with your comment. In the revised manuscript, we have better underlined this limit in the relevant paragraph 6. Specifically, we have cited two RCT where the comparison was fair and regarded only distal obstruction; then we have cited several meta-analyses where the obstruction site is mostly neither specified nor considered for subgroup analysis, predisposing to biases, and we have pointed this out.